# Temporal and spatial distributions and clustering features of soil-transmitted helminthiases on Hainan Island: A retrospective study from 2017–2023

Guangda Xu[1], Wen Zeng[2], Xiaomin Huang[2], Yongyan Tang[2], Yuchun Li[2]*

**1** Department of Epidemiology, Public Health School, Hainan Medical University, Haikou, Hainan, People's Republic of China, **2** Hainan Provincial Center for Disease Control and Prevention, Haikou, Hainan, People's Republic of China

* yuchunlee@126.com

## Abstract

### Objective

To summarize the epidemiological status and transmission dynamics of soil-transmitted helminthiases (STHs) in Hainan Island from 2017 to 2023, providing an evidence-based foundation for optimizing future control strategies.

### Methods

From 2017 to 2023, two mobile and one fixed surveillance sites were selected annually from 18 counties/cities in Hainan Island. Each site was divided into five geographical zones (east, west, south, north, and central), with one administrative village randomly chosen from each zone. At least 200 local residents aged ≥ 3 years were enrolled per village, totalling ≥ 1,000 participants per site. Fresh stool samples (≥ 30 g) were collected for modified Kato–Katz thick smear examination (two slides per sample) to detect hookworm, *Trichuris trichiura*, *Ascaris lumbricoides*, and *Enterobius vermicularis* eggs. Perianal cellophane tape swabs were used for *E. vermicularis* screening in children aged 3–9 years. Infection prevalence was calculated, and demographic differences were analysed by using the chi-square test. Temporal trends were assessed via Joinpoint regression analysis. ArcGIS 10.8 was used to construct a spatial database, perform kernel density estimation (KDE), and calculate global and local Moran's I indices for spatial aggregation analysis.

### Results

A total of 29,669 individuals from 128 administrative villages across 88 townships in 18 county-level divisions were monitored. The overall STH infection rate was 5.76% (1,710/29,669), with hookworms (3.47%), *A. lumbricoides* (0.88%), *T. trichiura*

**Data availability statement:** All relevant data are in the manuscript and its Supporting information files.

**Funding:** This work was supported by Joint Program on Health Science & Technology Innovation of Hainan Province(grant number WSJK2024MS167 to Y.L). The funders had no role in study design, data collection and analysis, decision to publish, or preparation of the manuscript.

**Competing interests:** The authors have declared that no competing interests exist.

(0.10%), and *E. vermicularis* (1.45%) as the dominant species. The infection rate showed a significant downward trend (AAPC = -18.99, $P < 0.05$). Regional differences were evident ($\chi^2 = 735.316$, $P < 0.001$), with the highest rate in Danzhou City (18.96%) and the lowest rate in Ding'an County (0.39%). Infection rates varied significantly by sex, age, ethnicity, occupation, and education level (all $P < 0.001$). Global spatial autocorrelation indicated clustering of *T. trichiura* infections (Moran's I = 0.137, Z = 1.983, $P < 0.05$). Local autocorrelation identified high–high clustering in Baisha County and low–low clustering in Haikou City and adjacent areas. KDE analysis revealed core high-density zones in Wuzhishan City and neighbouring counties, with secondary hotspots in Danzhou City and the Tunchang–Qionghai–Ding'an corridor.

## Conclusion

Although STH infection rates in Hainan Island have consistently declined, central–southern mountainous counties and vulnerable groups (e.g., children and older people) require sustained surveillance and targeted interventions to consolidate control efforts.

### Author summary

A consecutively 7-year study (2017–2023) tracked intestinal worm infections spread through soil in Hainan Island, China. Researchers tested nearly 30,000 people across 128 villages and found an overall infection rate of 5.76%. Hookworms were the most common type (3.47%), followed by *Enterobius vermicularis* (1.45%), *Ascaris lumbricoides* (0.88%), and *Trichuris trichiura* (0.10%). Infection rates decreased year by year during the study period and spatial analysis revealed hotspots concentrated in central-southern mountainous areas around the Wuzhishan City. Children (aged 3–10), older adults (over 61), kindergarteners, and those with little formal education still faced higher infection risks. Females were slightly more affected than males. The study recommends that significantly more efforts will be required to achieve the goal of controlling transmission and interrupting soil-transmitted nematodiasis. Targeted interventions for high-risk groups are essential, particularly for blocking the transmission of *E. vermicularis* infection in children.

## Introduction

*Ascaris lumbricoides*, *T. trichiura*, hookworms, and *Enterobius vermicularis* cause soil-transmitted helminthiases (STHs) [1], which are classified by the World Health Organization (WHO) as neglected tropical diseases. As of 2023, over 1.5 billion people globally were infected, with the WHO aiming to eliminate transmission by 2030 [2–4]. STH infection rates reflect regional socioeconomic development and public

health status [5]. Hainan Island in China's tropical zone provides ideal conditions for STH proliferation, remaining one of China's most endemic regions [6,7].

Post-1949, some sentinel surveys in Lingshui, Wanning, and Danzhou counties revealed STH infection rates >90%, with *A. lumbricoides* as the dominant species (63.8–100%), followed by hookworms (23.3–92.4%) and *T. trichiura* (31.8–90.8%), although islandwide data were historically incomplete [8]. Three national parasite surveys were conducted in 1986–1991, 2001–2004 and 2015 and showed a declining prevalence as follows: 94.7% (highest nationwide) [9], 56.2% (40.7% reduction) [10], and 11.8% (79.0% reduction) [11], respectively, with Hainan Island still ranking among China's top endemic zones. Since 2017, three sentinel sites in Hainan Island have been established for dynamic STH monitoring.

Spatial epidemiology, which integrates the geographic information system (GIS) and disease control, uses kernel density estimation (KDE) and spatial autocorrelation to characterize spatiotemporal patterns. KDE visualizes density distributions, while spatial autocorrelation (e.g., Moran's *I*) evaluates global clustering and local hotspots. These methods have been applied to infectious diseases (e.g., schistosomiasis, malaria, and echinococcosis) [12] and identified high-risk clusters in northwestern China [13], dengue hotspots in Fujian [14], and brucellosis aggregation in Jiangsu [15]. However, spatiotemporal analyses of STHs in Hainan remain limited. This study used spatial epidemiological methods to summarize and analyse STH infection characteristics during 2017–2023.

## Materials and methods

### Ethics statement

This study was approved by the Ethics Committee of the Hainan Provincial Center for Disease Control and Prevention (Approval No. 2021006). Written informed consent was obtained from all participants or legal guardians. The objectives, procedures and potential risks of this study were orally explained to all participants.

### Sample size calculation

Based on the population proportion sample size formula $N = \frac{Z^2 \times P \times (1-p)}{d^2}$, where P is the infection rate and d is the permissible error, with reference to the 11.8% provincial infection rate in the 2015 parasitic survey, a permissible error "d" of 0.015, and 95% confidence level (Z = 1.96), the baseline sample size was calculated to be n = 1,777; per field survey protocols requiring a 50% oversampling adjustment, the minimum annual target sample size was established at approximately 2666 participants.

### Surveillance site selection and study population

Following China's *National Surveillance Protocol for Clonorchiasis and Soil-transmitted Helminthiases (Trial)*, with minor modifications, Hainan selected three counties/cities annually as surveillance sites. By 2023, the two mobile and one fixed surveillance sites had covered all 18 counties/cities. Stratified random sampling was adopted. First, each site was divided into five geographical zones (east, west, south, north, and central), with one administrative village randomly selected per zone as the survey site. At least 200 permanent residents aged ≥3 years (across paediatric, adolescent, adult, and older groups) who had lived in the surveillance site for at least one year, had not taken anthelmintics, and had no diarrhea were enrolled per village. Mobile residents were excluded from sampling, ensuring ≥1,000 participants per site. The response rate was consistent with the National Survey on Human Parasitic Diseases, requiring no less than 85%.

### Survey methods and data sources

Data collection followed the protocol established by the Parasitic Disease Surveillance System of China. Household-based stool collection kits (labelled with unique IDs) were distributed, with standardized collection and transportation instructions; each participant was required to provide no less than 30 g of stool sample. For children aged 3–9 years,

perianal cellophane tape swabs were used for *E. vermicularis* screening. Demographic data were recorded by using unified forms. In accordance with China's National Health Industry Standard Intestinal Helminth Detection - *Modified Kato–Katz Thick Smear Method (WS/T 570–2017)*, stool samples underwent modified Kato–Katz thick smear examination (two slides per sample); experienced microscopic examination specialists tested the samples for hookworm, *A. lumbricoides*, *T. trichiura*, and *E. vermicularis* eggs; egg counting was performed, and infection rates were calculated by species. A slide was considered positive if at least one type of parasite egg was detected, and mixed infections were possible; infection rates were calculated by species. Quality control measures focused on the following: (i) provincial technical oversight by the Hainan CDC; (ii) microscopist training and certification; (iii) double-blind data entry and verification; and (iv) random reexamination of ≥ 10% positive and ≥ 5% negative slides.

### Data analysis

Data were collated with WPS Office 2016 and then analysed using SPSS 26.0 and Joinpoint 5.4.0. Differences in infection rates were compared using $\chi^2$ tests, and trend changes were analyzed using both Joinpoint regression analysis and trend $\chi^2$ tests ($P < 0.05$ defined as significant).

### Spatial analysis

A 1:1,000,000-scale vector map of Hainan's administrative divisions (WGS_1984 coordinate system, UTM Zone 49N projection) was downloaded from the Hainan Provincial Bureau of Surveying, Mapping and Geoinformation. County-level data were georeferenced to build a spatial database, with analysis and visualization performed by using ArcGIS 10.8. Case coordinates were extracted for KDE thematic mapping, with KDE values classified via the natural breaks method. Spatial autocorrelation analysis was conducted using the Global Moran's *I* and Anselin Local Moran's *I* tools in ArcGIS 10.8, with the spatial weight matrix set to INVERSE_DISTANCE and the distance threshold using the software-calculated minimum threshold. Global Moran's *I* (range: -1–1) was calculated for spatial autocorrelation analysis via the Z score test ($P < 0.05$ when $|Z| > 1.96$), indicating clustered (positive), dispersed (negative), or random (zero) distributions. Anselin Local Moran's *I* with LISA cluster maps identified high–high/low–low (positive association) and high–low/low–high (negative association) patterns. Results were stratified for epidemiological interpretation ($a = 0.05$).

## Results

### Overall infection profile

From 2017 to 2023, 29,669 individuals from 128 villages in 88 townships were screened. The overall prevalence of STH infections was 5.76% (1,710/29,669), with a significantly higher positivity rate in dry season samples (8.6% [444/5,156]) compared to rainy season samples (5.2% [1,266/24,513]) ($\chi^2 = 93.177$, $P < 0.001$). Regarding the infection rates among different species, hookworms were the dominant species (3.47% [1,030/29,669]), followed by *E. vermicularis* (1.45% [430/29,669]), *T. trichiura* (0.88% [260/29,669]), and *A. lumbricoides* (0.10% [30/29,669]). *E. vermicularis* was detected in 113 cases via the Kato–Katz method (infection rate: 0.38% [113/29,669]) and in 326 cases via perianal swabs in children (infection rate: 6.47% [326/5,037]), which showed a significant difference ($\chi^2 = 1279.234$, $P < 0.001$).

### Temporal distribution

The annual infection rates showed significant differences ($\chi^2 = 699.336$, $P < 0.001$), declining from 14.73% in 2017 to 4.07% in 2023. Joinpoint regression analysis confirmed a consistent annual downward trend in STH and hookworm infection rates (AAPC = -18.99%, $P < 0.05$; AAPC = -32.61%, $P < 0.05$). However, the trend changes in infection rates of other species were not significant: *T. trichiura* (AAPC = -23.10%, $P = 0.13$) and *E. vermicularis* (AAPC = 4.94%, $P = 0.71$). For *A. lumbricoides,* due to extremely low infection rates over the years, Joinpoint trend analysis could not be performed (Table 1 and Fig 1A–B).

**Table 1. The infection of soil-transmitted helminthiase in Hainan Island in 2017-2023.**

| Year | No. surveyed | Soil-transmitted helminthiase | | Hookworm | | *Ascaris lumbricoides* | | *Trichuris trichiura* | | *Enterobius vermicularis* | |
|---|---|---|---|---|---|---|---|---|---|---|---|
| | | No. infected | Overall infection rate/% | No. infected | Infection rate/% | No. infected | Infection rate/% | No. infected | Infection rate/% | No. infected | Infection rate/% |
| 2017 | 3028 | 446 | 14.73 | 312 | 10.30 | 24 | 0.79 | 98 | 3.24 | 42 | 1.39 |
| 2018 | 3095 | 243 | 7.85 | 203 | 6.56 | 0 | 0 | 11 | 0.36 | 29 | 0.94 |
| 2019 | 3227 | 147 | 4.56 | 89 | 2.76 | 0 | 0 | 19 | 0.59 | 39 | 1.21 |
| 2020 | 3141 | 274 | 8.72 | 166 | 5.28 | 1 | 0.03 | 32 | 1.02 | 77 | 2.45 |
| 2021 | 4697 | 165 | 3.51 | 100 | 2.13 | 0 | 0 | 51 | 1.09 | 20 | 0.43 |
| 2022 | 8304 | 265 | 3.19 | 138 | 1.66 | 5 | 0.06 | 44 | 0.53 | 79 | 0.95 |
| 2023 | 4177 | 170 | 4.07 | 22 | 0.53 | 0 | 0 | 5 | 0.12 | 144 | 3.45 |
| $\chi^2$ | | 699.336 | | 760.065 | | 79.401* | | 249.019 | | 194.789 | |
| P | | <0.001 | | <0.001 | | <0.001 | | <0.001 | | <0.001 | |

Note: *, The results are obtained by Fisher's Exact Test.

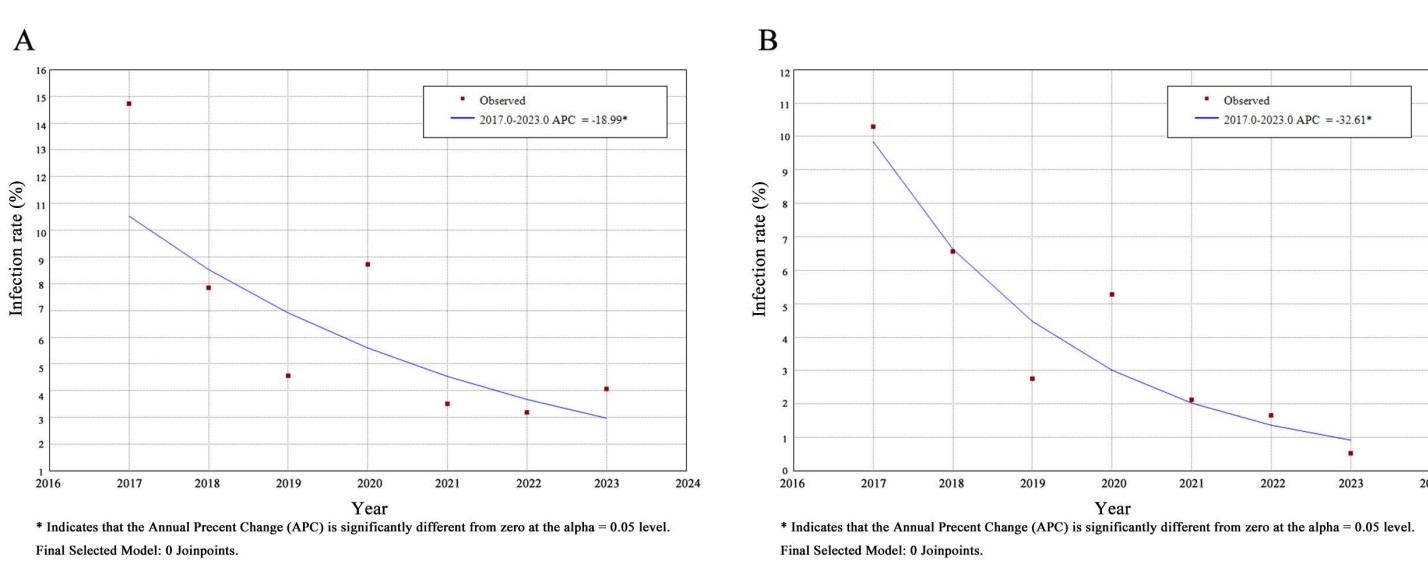

* Indicates that the Annual Precent Change (APC) is significantly different from zero at the alpha = 0.05 level.
Final Selected Model: 0 Joinpoints.

**Fig 1. Joinpoint regression analysis on the changing trends of annual incidence rates of STH and Hookworm in Hainan Island from 2017 to 2023.** Note: A: STH; B: Hookworm.

## Geographical distribution

From 2017 to 2023, STH infections were detected in all 18 cities and counties of Hainan Island, with infection rates in 7 cities and counties exceeding the average level and the county-level infection rates ranging from 0.39% (2/512) in Ding'an to 18.96% (193/1,018) in Danzhou. There were significant differences in STH infection rates among different regions ($\chi^2$ = 735.316, P < 0.001). Marked regional disparities existed ($\chi^2$ = 735.316, P < 0.001), with Danzhou City (18.96%) and Ding'an County (0.39%) showing the highest and lowest rates, respectively. The species-specific hotspots included Danzhou City for hookworms (11.00%), *A. lumbricoides* (2.06%), and *T. trichiura* (7.56%) and Chengmai County for *E. vermicularis* (8.24%) (Table 2).

**Table 2. Regional distribution of soil-transmitted helminthiase infection in Hainan Island in 2017-2023.**

| Cities (counties) | No. surveyed | Soil-transmitted helminthiase | | Hookworm | | *Ascaris lumbricoides* | | *Trichuris trichiura* | | *Enterobius vermicularis* | |
|---|---|---|---|---|---|---|---|---|---|---|---|
| | | No. infected | Overall infection rate/% | No. infected | Infection rate/% | No. infected | Infection rate/% | No. infected | Infection rate/% | No. infected | Infection rate/% |
| Baisha | 1697 | 88 | 5.19 | 41 | 2.42 | 3 | 0.18 | 30 | 1.77 | 18 | 1.06 |
| Baoting | 1009 | 30 | 2.97 | 25 | 2.48 | 2 | 0.20 | 0 | 0.00 | 3 | 0.30 |
| Changjiang | 3130 | 156 | 4.98 | 12 | 0.38 | 0 | 0.00 | 27 | 0.86 | 119 | 3.80 |
| Chengmai | 1104 | 99 | 8.97 | 8 | 0.72 | 0 | 0.00 | 0 | 0.00 | 91 | 8.24 |
| Danzhou | 1018 | 193 | 18.96 | 112 | 11.00 | 21 | 2.06 | 77 | 7.56 | 7 | 0.69 |
| Ding'an | 512 | 2 | 0.39 | 0 | 0.00 | 1 | 0.20 | 0 | 0.00 | 1 | 0.20 |
| Dongfang | 1502 | 14 | 0.93 | 6 | 0.40 | 0 | 0.00 | 5 | 0.33 | 4 | 0.27 |
| Haikou | 495 | 5 | 1.01 | 2 | 0.40 | 0 | 0.00 | 2 | 0.40 | 1 | 0.20 |
| Ledong | 508 | 10 | 1.97 | 5 | 0.98 | 0 | 0.00 | 3 | 0.59 | 2 | 0.39 |
| Lin'gao | 1091 | 55 | 5.04 | 16 | 1.47 | 0 | 0.00 | 4 | 0.37 | 35 | 3.21 |
| Lingshui | 1589 | 76 | 4.78 | 49 | 3.08 | 1 | 0.06 | 22 | 1.38 | 4 | 0.25 |
| Qionghai | 2063 | 160 | 7.76 | 142 | 6.88 | 0 | 0.00 | 10 | 0.48 | 12 | 0.58 |
| Qiongzhong | 3133 | 50 | 1.60 | 39 | 1.24 | 0 | 0.00 | 7 | 0.22 | 4 | 0.13 |
| Sanya | 2073 | 122 | 5.89 | 82 | 3.96 | 1 | 0.05 | 23 | 1.11 | 17 | 0.82 |
| Tunchang | 1553 | 92 | 5.92 | 40 | 2.58 | 0 | 0.00 | 5 | 0.32 | 47 | 3.03 |
| Wanning | 1513 | 41 | 2.71 | 32 | 2.12 | 1 | 0.07 | 5 | 0.33 | 3 | 0.20 |
| Wenchang | 1015 | 82 | 8.08 | 76 | 7.49 | 0 | 0.00 | 6 | 0.59 | 0 | 0.00 |
| Wuzhishan | 4664 | 435 | 9.33 | 343 | 7.35 | 0 | 0.00 | 34 | 0.73 | 62 | 1.33 |
| Total | 29669 | 1710 | 5.76 | 1030 | 3.47 | 30 | 0.10 | 260 | 0.88 | 430 | 1.45 |

## Demographic distribution

Infection rates varied significantly between 2017 and 2023 by demographics. Sex distribution analysis indicated a higher infection rate in females (6.49%, 976/15,049) than in males (5.02%, 734/14,620; $\chi^2 = 29.552$, $P < 0.001$). STH infections were found across all age groups, with the highest infection rate observed in children aged 3–10 years (9.00%, 503/5586), followed by individuals over 61 years old (7.61%, 453/5,954), while adolescents aged 11–20 years had the lowest rate (3.16%, 76/2,403). While significant age-related variation was evident ($\chi^2 = 244.854$, $P < 0.001$), no linear age-dependent trend was detected ($\chi^2_{trend} = 2.395$, $P = 0.122$).

Ethnic group-based analysis showed a higher STH prevalence in the Han Chinese (6.33%, 781/12,336) versus the Li Chinese (5.59%, 873/15,625), and the differences in STH infection rates among different ethnic groups were statistically significant ($\chi^2 = 27.955$, $P < 0.001$). Occupational and educational gradients further highlighted risk hierarchies, with kindergarten children having the highest infection rate (9.32%, 184/1,974), followed by nonenrolled children (8.55%, 37/433), while the unclassified other occupational groups showed the lowest infection rate (2.92%, 7/240). The differences among various occupational groups were statistically significant ($\chi^2 = 85.225$, $P < 0.001$). Education level-based analysis showed peaks among preschool children (9.21%, 224/2,432) and illiterate/semi-illiterate individuals (8.01%, 140/1,746), in stark contrast to college-educated groups (2.46%, 14/568). The differences in STH infection rates among participants with different education levels were statistically significant ($\chi^2 = 154.902$, $P < 0.001$).

The species-specific analyses showed similar demographic differences. The hookworm infection rates showed significant differences across the sexes, ages, ethnic groups, occupations, and education levels. The *A. lumbricoides* infection rates demonstrated statistically significant variations among different ages, ethnic groups, occupations, and education

levels, but no significant difference was observed between the sexes ($\chi^2 = 0.006$, $P > 0.05$). The *T. trichiura* infection rates exhibited significant differences among various ages, ethnic groups, and occupations, while no significant variations were found between the sexes or education levels ($\chi^2 = 0.019$ and $\chi^2 = 7.046$, respectively; both $P > 0.05$). The *E. vermicularis* infection rates displayed significant differences across ages, ethnic groups, occupations, and education levels, but no significant difference was detected between the sexes ($\chi^2 = 1.12$, $P > 0.05$; Table 3).

## Spatial analysis

KDE identified a core high-density zone in Wuzhishan City and central–southern counties, with secondary hotspots in Danzhou City and the Tunchang–Qionghai–Ding'an corridor (Figs 2 and 3A). Global autocorrelation showed clustering only for *T. trichiura* infection (Moran's $I = 0.14$, $Z = 1.98$, $P < 0.05$), while those caused by the other species showed random distributions (Moran's $I = 0.02$–$0.15$, all $Z < 1.96$, $P > 0.05$; Table 4). Local spatial autocorrelation identified distinct aggregation patterns across Hainan Island, with STH, hookworm, *T. trichiura*, *A. lumbricoides*, and *E. vermicularis* infections clustered in 6, 3, 8, 1, and 5 counties/cities, respectively. Specifically, STHs exhibited high–high clustering in Lin'gao County, high–low clustering in Wenchang City, and low–low clustering in Xiuying, Longhua, Qiongshan, and Meilan Districts; hookworm infection showed high–low aggregation in Wenchang City and low–low aggregation in Xiuying and Longhua Districts; *T. trichiura* infection displayed high–high clustering in Baisha County and low–low clustering in Xiuying, Longhua, Qiongshan, Meilan, Chengmai, Ding'an, and Tunchang Counties; *A. lumbricoides* infection manifested high–low aggregation in Ding'an County, while *E. vermicularis* infection demonstrated high–high clustering in Lingao County, low–high clustering in Dongfang City, and low–low clustering in Meilan, Qiongshan, and Wenchang (Fig 3B–3F and Table 5).

## Discussion

The geographical position of Hainan Island at China's southern extremity endows it with a tropical maritime monsoon climate characterized by mean annual temperatures exceeding 23.0 °C and substantially elevated precipitation and soil moisture retention relative to similar latitudes. Coupled with widespread traditional cultivation practices in rural areas, these conditions collectively establish multidimensional ecological parameters conducive to STH life cycle completion [6,16]. Historical data from China's national parasite surveys (1986–1991, 2001–2004, and 2015) revealed STH infection rates of 94.7%, 56.2%, and 11.8%, respectively, across Hainan [10,17,18]. Contemporary surveillance, however, has revealed persistently elevated endemicity, despite declining trajectories. Thus, national STH monitoring in 2019 recorded a 3.28% infection rate (105/3,205), with Hainan ranking fourth nationally [19]; subsequent surveillance in 2020 revealed a 6.34% prevalence (199/3,141), positioning Hainan first among all provinces [20]; and in 2021, the rate declined to 3.13% (131/4,184) while maintaining the third-highest national incidence [21].

By leveraging Hainan Island's dynamic surveillance network (2017–2023), this study provides the first systematic analysis of STH infection trends post-2015. An overall prevalence of 5.8% was documented—representing a 50.8% reduction from that during the third national survey—with Joinpoint regression analysis confirming significant annual declines in both the annual average infection rates of STH and hookworm, paralleling epidemiological patterns in STH high-prevalence provinces such as Sichuan, Yunnan, Guizhou, and Jiangxi, mainly attributed to improvements in public health conditions, changes in lifestyle, and the advancement of targeted prevention and control measures [22–25]. Hookworms were the dominant species (mean prevalence: 3.5%), which is consistent with historical Hainan data [17,26] and potentially attributable to inadequate farmer awareness of transmission pathways, insufficient protective measures during agricultural activities, and prevalent barefoot farming practices [27]. Additionally, infection rates in samples collected during the dry season were higher than those in the rainy season—consistent with studies from Ethiopia and Brazil. This may be attributed to the drought resistance of some STH eggs, the peak of agricultural harvesting/planting in the dry season, and the combined scouring effect of rainwater in the rainy season [28–30]. Geospatial analysis revealed unprecedented clustering in the west, with Danzhou city exhibiting the highest infection rate, diverging from prior central-region predominance [8].

**Table 3. Distribution of soil-transmitted helminthiase infection in different population of Hainan Island in 2017-2023.**

| Feature | No. surveyed | Soil-transmitted helminthiase | | Hookworm | | Ascaris lumbricoides | | Trichuris trichiura | | Enterobius vermicularis | |
|---|---|---|---|---|---|---|---|---|---|---|---|
| | | No. infected | Overall infection rate/% | No. infected | Infection rate/% | No. infected | Infection rate/% | No. infected | Infection rate/% | No. infected | Infection rate/% |
| **Gender** | | | | | | | | | | | |
| Male | 14620 | 734 | 5.02 | 408 | 2.79 | 15 | 0.10 | 127 | 0.87 | 201 | 1.37 |
| Female | 15049 | 976 | 6.49 | 622 | 4.13 | 15 | 0.10 | 133 | 0.88 | 229 | 1.52 |
| $\chi^2$ | | 29.552 | | 39.881 | | 0.006 | | 0.019 | | 1.12 | |
| $P$ | | <0.001 | | <0.001 | | 0.937 | | 0.889 | | 0.29 | |
| **Age group** | | | | | | | | | | | |
| 3~10 | 5586 | 503 | 9.00 | 36 | 0.64 | 21 | 0.38 | 76 | 1.36 | 386 | 6.91 |
| 11~20 | 2403 | 76 | 3.16 | 43 | 1.79 | 0 | 0.00 | 32 | 1.33 | 8 | 0.33 |
| 21~30 | 2226 | 92 | 4.13 | 63 | 2.83 | 2 | 0.09 | 24 | 1.08 | 7 | 0.31 |
| 31~40 | 4256 | 154 | 3.62 | 127 | 2.98 | 1 | 0.02 | 23 | 0.54 | 5 | 0.12 |
| 41~50 | 4369 | 187 | 4.28 | 161 | 3.69 | 2 | 0.05 | 20 | 0.46 | 9 | 0.21 |
| 51~60 | 4875 | 245 | 5.03 | 199 | 4.08 | 1 | 0.02 | 38 | 0.78 | 9 | 0.18 |
| 61~70 | 3454 | 261 | 7.56 | 234 | 6.77 | 2 | 0.06 | 26 | 0.75 | 2 | 0.06 |
| ≥71 | 2500 | 192 | 7.68 | 167 | 6.68 | 1 | 0.04 | 21 | 0.84 | 4 | 0.16 |
| $\chi^2$ | | 244.854 | | 354.541 | | 35.245* | | 37.369 | | 1437.932 | |
| $P$ | | <0.001 | | <0.001 | | <0.001 | | <0.001 | | <0.001 | |
| **Nationality** | | | | | | | | | | | |
| Han | 12336 | 781 | 6.33 | 434 | 3.52 | 25 | 0.20 | 140 | 1.13 | 206 | 1.67 |
| Li | 15625 | 873 | 5.59 | 562 | 3.60 | 5 | 0.03 | 107 | 0.68 | 210 | 1.34 |
| Miao | 1646 | 55 | 3.34 | 33 | 2.00 | 0 | 0.00 | 13 | 0.79 | 14 | 0.85 |
| Others | 62 | 1 | 1.61 | 1 | 1.61 | 0 | 0.00 | 0 | 0.00 | 0 | 0.00 |
| $\chi^2$ | | 27.955 | | 12.083 | | 21.445* | | 15.762* | | 9.744* | |
| $P$ | | <0.001 | | 0.007 | | <0.001 | | 0.001 | | 0.018 | |
| **Occupation** | | | | | | | | | | | |
| Farmer | 18950 | 1038 | 5.48 | 887 | 4.68 | 7 | 0.04 | 132 | 0.70 | 28 | 0.15 |
| Migrant workers and workers | 981 | 31 | 3.16 | 23 | 2.34 | 2 | 0.20 | 9 | 0.92 | 0 | 0.00 |
| Students | 5378 | 342 | 6.36 | 48 | 0.89 | 8 | 0.15 | 78 | 1.45 | 219 | 4.07 |
| Scattered children | 433 | 37 | 8.55 | 3 | 0.69 | 3 | 0.69 | 11 | 2.54 | 22 | 5.08 |
| Nursery children | 1974 | 184 | 9.32 | 15 | 0.76 | 10 | 0.51 | 14 | 0.71 | 153 | 7.75 |
| Medical Professionals, Administrative Personnel and Educators | 260 | 10 | 3.85 | 5 | 1.92 | 0 | 0.00 | 4 | 1.54 | 1 | 0.38 |
| Retiree | 642 | 34 | 5.30 | 26 | 4.05 | 0 | 0.00 | 7 | 1.09 | 1 | 0.16 |
| Unemployed | 811 | 27 | 3.33 | 18 | 2.22 | 0 | 0.00 | 5 | 0.62 | 4 | 0.49 |
| Others | 240 | 7 | 2.92 | 5 | 2.08 | 0 | 0.00 | 0 | 0.00 | 2 | 0.83 |
| $\chi^2$ | | 85.225 | | 254.11 | | 38.553* | | 46.309 | | 1102.381 | |
| $P$ | | <0.001 | | <0.001 | | <0.001 | | <0.001 | | <0.001 | |
| **Education level** | | | | | | | | | | | |
| illiterate/semiliterate | 1746 | 140 | 8.02 | 123 | 7.04 | 2 | 0.11 | 17 | 0.97 | 3 | 0.17 |
| Preschoolers | 2432 | 224 | 9.21 | 19 | 0.78 | 14 | 0.58 | 26 | 1.07 | 176 | 7.24 |
| Primary school | 9192 | 633 | 6.89 | 322 | 3.50 | 9 | 0.10 | 95 | 1.03 | 219 | 2.38 |

*(Continued)*

**Table 3.** (Continued)

| Feature | No. surveyed | Soil-transmitted helminthiase | | Hookworm | | *Ascaris lumbricoides* | | *Trichuris trichiura* | | *Enterobius vermicularis* | |
|---|---|---|---|---|---|---|---|---|---|---|---|
| | | No. infected | Overall infection rate/% | No. infected | Infection rate/% | No. infected | Infection rate/% | No. infected | Infection rate/% | No. infected | Infection rate/% |
| Junior high school | 13370 | 610 | 4.56 | 490 | 3.66 | 5 | 0.04 | 100 | 0.75 | 27 | 0.20 |
| High school/secondary school | 2361 | 89 | 3.77 | 70 | 2.96 | 0 | 0.00 | 17 | 0.72 | 2 | 0.08 |
| College and above | 568 | 14 | 2.46 | 6 | 1.06 | 0 | 0.00 | 5 | 0.88 | 3 | 0.53 |
| $\chi^2$ | | 154.902 | | 132.261 | | 35.719* | | 7.046 | | 826.122 | |
| $P$ | | <0.001 | | <0.001 | | <0.001 | | 0.217 | | <0.001 | |

Note: *, The results are obtained by Fisher's Exact Test.

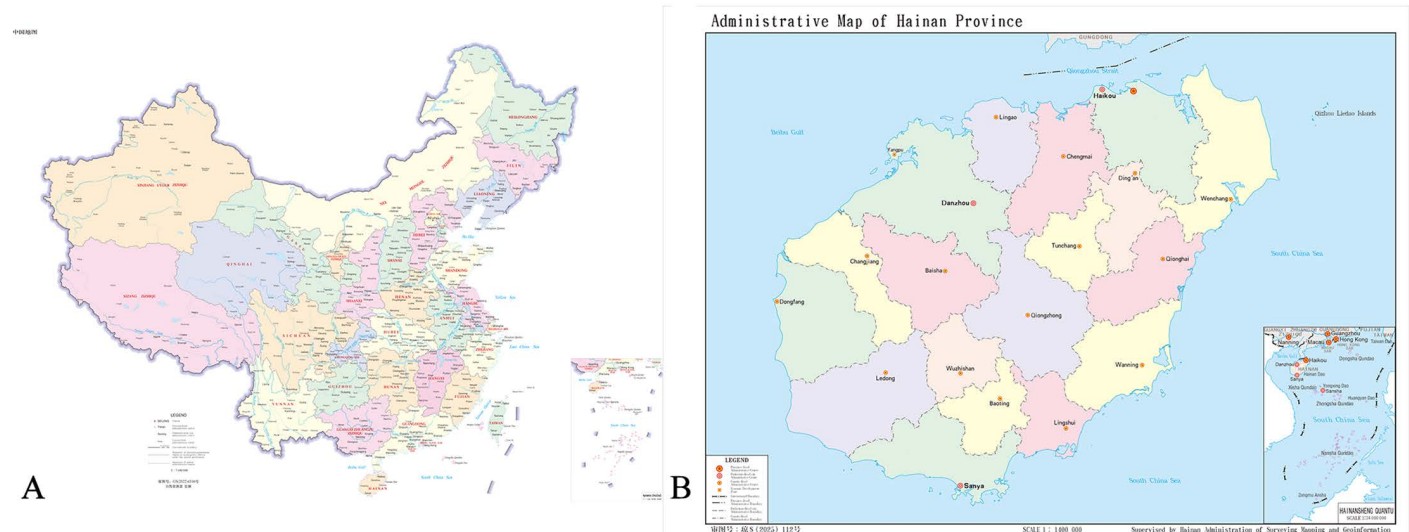

**Fig 2. Location and geographic maps in Hainan, China.** (A) Map of China (GS(2022)4316); (B) Map of Hainan island (琼S(2024)100). These maps were created based on publicly provided base maps from China Map Press and Hainan Administration of Surveying, Mapping and Geoinformation, with the base map for Panel A available at http://bzdt.ch.mnr.gov.cn/browse.html?picId=%224o28b0625501ad13015501ad2bfc0690%22 and that for Panel B at http://chj.i0898.org/sjkf/hndt/hnsdt/zqb/202208/t20220825_3390475.html.

Demographic stratification revealed significant disparities: elevated female versus male infection rates (conforming with national trends [31], potentially reflecting gender-based agricultural exposure differentials [32,33]); higher Han ethnicity prevalence versus historical minority dominance [10,17,18], likely attributable to improvements in sanitation infrastructure expanded safe-water coverage in minority regions. According to the Hainan Statistical Yearbook from 2017 to 2023, with the background of poverty alleviation and beautiful countryside construction, rural sanitary toilet coverage in minority regions (including Li, Miao, and other ethnic groups) increased from 86.3% to 99.3%, while safe drinking water coverage rose from 84.5% to 94.0% [8,34]; and peak infections among children aged 3–10 years, kindergarten/nonenrolled children, and preschoolers—mirroring findings in most tropical Asian regions such as Malaysia, Thailand, and the Philippines [35] and implicating institutional cross-transmission, suboptimal hygiene practices, and limited health literacy among children [36]. These groups are thus designated as priority intervention targets.

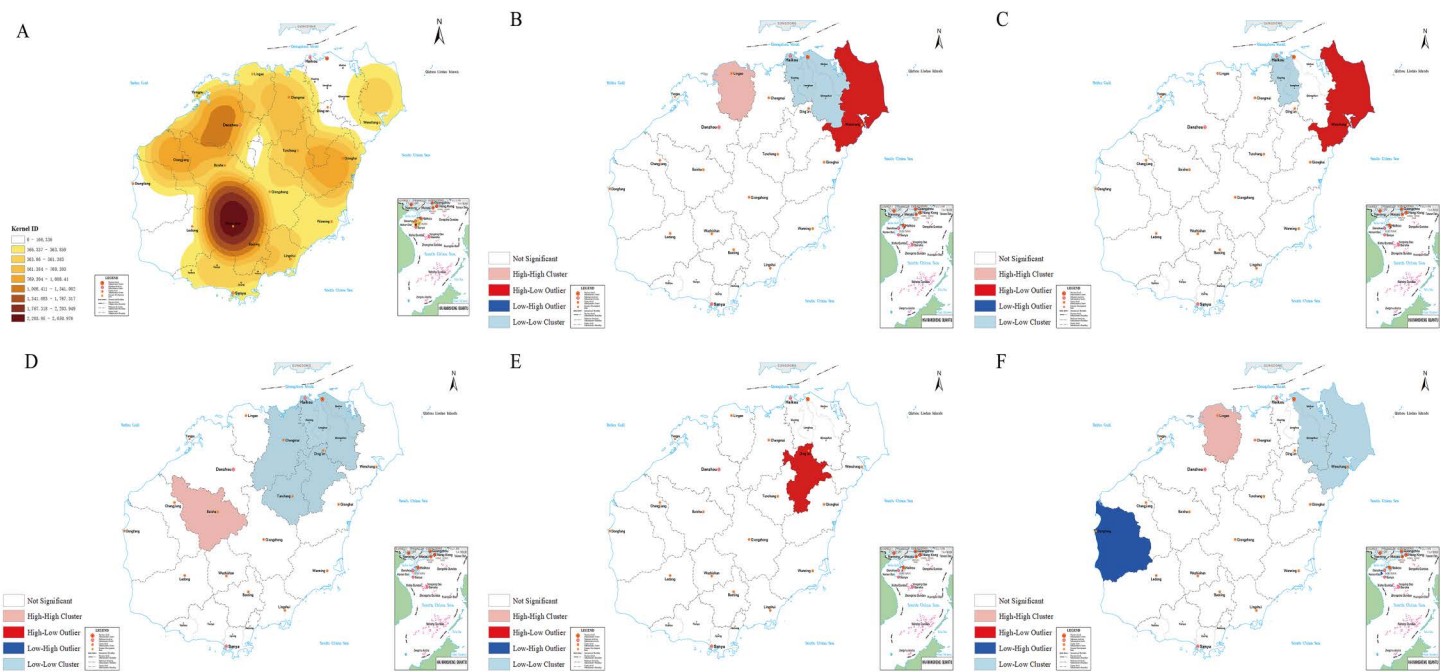

**Fig 3. Local spatial autocorrelation analysis distribution map of soil-transmitted nematode and various parasite infection rates in Hainan Island from 2017 to 2023 Drawing Audit Number: 琼S(2025)249.** Note: A: Estimation of kernel density of STH; B: Soil-transmitted helminthiase; C: Hookworm; D: Trichuris trichiura; E: Ascaris lumbricoides; F: Enterobius vermicularis. These maps were produced based on the publicly provided base map from Hainan Administration of Surveying, Mapping and Geoinformation, where the base map for Panels A–F is available at http://chj.i0898.org/sjkf/hndt/hnsdt/zqb/202208/t20220825_3390475.html.

**Table 4. Results of global spatial autocorrelation analysis of soil-transmitted helminthiase infection in Hainan Island.**

| Species | Moran's *I* | Variance | *Z* | *P* |
|---|---|---|---|---|
| Soil-transmitted helminthiase | 0.148 | 0.021 | 1.313 | 0.189 |
| Hookworm | 0.110 | 0.024 | 0.986 | 0.324 |
| *Trichuris trichiura* | 0.137 | 0.008 | 1.983 | 0.047 |
| *Ascaris lumbricoides* | 0.022 | 0.004 | 1.100 | 0.271 |
| *Enterobius vermicularis* | 0.151 | 0.016 | 1.543 | 0.123 |

**Table 5. Results of local spatial autocorrelation analysis of soil-transmitted helminthiase and various parasite infection rates in Hainan Island.**

| Type of clusters | Soil-transmitted helminthiase | Hookworm | *Trichuris trichiura* | *Ascaris lumbricoides* | *Enterobius vermicularis* |
|---|---|---|---|---|---|
| High-high cluster | Lin'gao | / | Baisha | / | Lin'gao |
| High-low cluster | Wenchang | Wenchang | / | Ding'an | / |
| Low-high cluster | / | / | / | / | Dongfang |
| Low-low cluster | Xiuying, Longhua, Qiongshan, Meilan | Xiuying, Longhua | Xiuying, Longhua, Qiongshan, Meilan, Chengmai, Tunchang | / | Meilan, Qiongshan, Wenchang |

Recent advancements in spatial statistical analysis have been extensively applied to elucidate the spatiotemporal distribution and clustering patterns of infectious diseases, enabling the identification of high-risk foci and dynamic transmission trends [37]. In this investigation, KDE revealed, for the first time, a core high-density zone of STH infections centred on Wuzhishan City and its central–southern periphery, with secondary high-density zones emerging in Danzhou City and the Tunchang–Qionghai–Ding'an corridor. Compared with other areas, Wuzhishan City, which is located in a mountainous central region, has ecological conditions—including an annual average temperature of 23.7°C, annual rainfall of 1483.1 mm, concurrent rain and heat periods, and a mountainous agricultural model (where climate is sitable for economic crops such as betel nut or rubber tree growing and its require field fertilization, increasing opportunities for soil contact)—that are more conducive to STH propagation than in other cities/counties. Wuzhishan City and its central-southern periphery remain the primary hotspot for STH infections in Hainan, requiring sustained intervention efforts, although its consistently ranks among the highest endemic regions in historical surveys [8]. The global spatial autocorrelation analysis indicated spatial clustering solely for *T. trichiura* infections (Moran's $I = 0.14$, $Z = 1.98$, $P < 0.05$), whereas STH, hookworm, *A. lumbricoides*, and *E. vermicularis* infections exhibited random spatial distributions. Local spatial autocorrelation further delineated *T. trichiura* hotspots: a high–high cluster in Baisha County and a low–low cluster concentrated in Haikou city and adjacent counties—patterns potentially driven by ecological factors, regional economic development levels, and agricultural practices. The urban infrastructure, robust public health systems, and heightened health literacy in Haikou City collectively underpin its low infection aggregation, whereas agrarian–forestry economy, optimal STH developmental conditions, and constrained public health resources of Baisha County critically sustain its high prevalence status.

Despite confirming a sustained decrease in STH infection rates across Hainan Island, this study revealed that transmission control thresholds remained unmet. According to *China's Criteria for Transmission Control and Interruption of Soil-transmitted Nematodiasis (WS/T 629–2018)*, achieving transmission control requires maintaining a below-1% prevalence for three consecutive years [38]. However, surveillance data from 2021–2023 demonstrate persistent exceedance (3.51% (2021), 3.19% (2022), and 4.07% (2023)), indicating that Hainan necessitates an extended consolidation phase compared with other provinces. To achieve the goal for Transmission Control and Interruption of Soil-transmitted Nematodiasis, targeted interventions for high-risk groups are required. Similar to the STH intervention programs for children in Philippines, in response to the relatively high *E. vermicularis* infection rate among children, multi-stakeholder collaboration could be attempted: providing regular free *E. vermicularis* testing for children aged 3–10 years in conjunction with other public health programs targeting children, distributing targeted free anthelmintics for children in schools or kindergartens, and combining handwashing after entering schools or kindergartens with anal washing via smart toilets [39]. Additionally, the groups of agricultural workers, females, and individuals over 60 years of age have a high degree of overlap. For these groups, health departments should collaborate with rural agricultural departments to conduct health education and drug-based deworming, and integrate stool testing services with basic public health services.

This study also has certain limitations. Methodologically, the sampling-based survey design introduces potential selection bias; additionally, the large disparity in sample collection between the rainy and dry seasons may lead to an underestimation of actual infection rates to some extent. Furthermore, hookworms, as the dominant species in Hainan, have relatively lower detection sensitivity with the modified Kato-Katz thick smear method compared to quantitative polymerase chain reaction (qPCR) due to their biological characteristics; manual testing also exhibits fluctuations in detection capability, which may underestimate actual infection rates to some degree [39]. From a statistical perspective, spatial statistical methods require a foundation of global data to better explain the distribution patterns of the studied attribute values within a region; spatial autocorrelation analysis applied to sampled data may still yield skewed spatial interpretations, and spatial interpolation techniques may average out local outliers, thereby masking small-scale high-risk areas. Consequently, divergence between the reported STH prevalence and actual provincial infection levels cannot be ruled out, though this study still reflects infection characteristics to a certain extent. This also provides insights for subsequent research: for example, using more sensitive rapid detection test strips or automatic stool analysis machines for automated testing to reduce

fluctuations in detection capability and improve detection sensitivity and specificity; in addition, exploring more appropriate and accurate statistical models to better utilize sampled data for interpretation.

## Supporting information

**S1 Data. Complete raw data underlying the analytical results.**
(XLSX)

**S2 Data. Annual Sanitary Toilet Coverage Data (2017–2023) in Hainan.**
(XLSX)

**S3 Data. Raw data for Joinpoint regression analysis.**
(CSV)

## Acknowledgments

We would like to thank all the participants in this study. We also thank all the staff of the Center of Diseases Control and Prevention in 18 counties who participated in stool sample collection and examination, as well as data sorting. Jihao Wang*, Guosheng Wang*, Guoyi Wang*, Jian Wang*, Datian Zhan*, Anjun Shi*, Qing Zhu*, Wuhan Liu*, Haishan Li*, Hai Zhang*, Zaichun Zheng*, Chaozhi Hong*, Huanzhi Xu*, Shaoling Huang*, Ribiao Zeng*, Deyuan Zeng*, Jiqian Cai*. * authors listed by order of Chinese characters.

## Author contributions

**Conceptualization:** Yuchun Li.

**Data curation:** Wen Zeng.

**Formal analysis:** Guangda Xu.

**Funding acquisition:** Yuchun Li.

**Investigation:** Xiaomin Huang.

**Project administration:** Wen Zeng.

**Resources:** Xiaomin Huang, Yongyan Tang.

**Writing – original draft:** Guangda Xu.

**Writing – review & editing:** Yuchun Li.

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
