## [Decision Letter · Decision Letter 0]

26 Sep 2025

PNTD-D-25-01212

Temporal and spatial distributions and clustering features of soil-transmitted helminthiases on Hainan Island: a retrospective study from 2017–2023

Dear Dr. Li,

Thank you for submitting your manuscript to PLOS Neglected Tropical Diseases. After careful consideration, we feel that it has merit but does not fully meet PLOS Neglected Tropical Diseases's publication criteria as it currently stands. Therefore, we invite you to submit a revised version of the manuscript that addresses the points raised during the review process.

Please submit your revised manuscript within 60 days Nov 25 2025 11:59PM. If you will need more time than this to complete your revisions, please reply to this message or contact the journal office at plosntds@plos.org. Please include the following items when submitting your revised manuscript:

We look forward to receiving your revised manuscript.

Kind regards,

María Victoria Periago

Academic Editor

Krystyna Cwiklinski

Section Editor

Shaden Kamhawi

co-Editor-in-Chief

Paul Brindley

co-Editor-in-Chief

**Journal Requirements:**

At this stage, the following Authors/Authors require contributions: Guangda Xu, Wen Zeng, Xiaomin Huang, Yongyan Tang, and Yuchun Li. Please ensure that the full contributions of each author are acknowledged in the "Add/Edit/Remove Authors" section of our submission form.

Potential Copyright Issues:

- Figures 1 and 2. Please (a) provide a direct link to the base layer of the map (i.e., the country or region border shape) and ensure this is also included in the figure legend; and (b) provide a link to the terms of use / license information for the base layer image or shapefile. We cannot publish proprietary or copyrighted maps (e.g. Google Maps, Mapquest) and the terms of use for your map base layer must be compatible with our CC BY 4.0 license.

6) Please ensure that the funders and grant numbers match between the Financial Disclosure field and the Funding Information tab in your submission form. Note that the funders must be provided in the same order in both places as well.

**Reviewers' Comments:**

Reviewer's Responses to Questions

**Results**

-Does the analysis presented match the analysis plan?

-Are the results clearly and completely presented?

-Are the figures (Tables, Images) of sufficient quality for clarity?

Reviewer #1: The analyses presented are appropriate. The data align with the analysis plan. Additionally, the results are well presented, and the tables are of high quality.

Reviewer #2: -Does the analysis presented match the analysis plan? Yes

-Are the results clearly and completely presented? Yes

-Are the figures (Tables, Images) of sufficient quality for clarity? Yes

Reviewer #3: (No Response)

**Conclusions:**

-Are the conclusions supported by the data presented?

-Are the limitations of analysis clearly described?

-Do the authors discuss how these data can be helpful to advance our understanding of the topic under study?

-Is public health relevance addressed?

Reviewer #1: 1. Yes, the conclusions appear to be supported by the data.

2. Yes, the specifications of the analyses are provided when the authors highlight that the work was carried out by sampling, noting that this approach may introduce selection bias, and that the spatial autocorrelation analysis applied to the data can produce spatial distortions.

3. Yes, the authors discuss that there was a reduction in STH infection rates on Hainan Island but transmission control remained unachieved.

4. Yes, the public health relevance is addressed. The discussion highlights that Haikou City's urban infrastructure, robust public health systems, and high health literacy contribute to its low infection clustering, whereas Baisha County's agrarian-forestry economy, favorable conditions for STH development, and limited public health resources contribute to its high prevalence.

Reviewer #2: -Are the conclusions supported by the data presented? Yes

-Are the limitations of analysis clearly described? Yes

-Do the authors discuss how these data can be helpful to advance our understanding of the topic under study? Not completely

Reviewer #3: (No Response)

**Editorial and Data Presentation Modifications?**

Reviewer #1: (No Response)

Reviewer #2: See in the Summary

Reviewer #3: (No Response)

**Summary and General Comments:**

Reviewer #1: The study, "Temporal and spatial distributions and clustering features of soil-transmitted helminthiases on Hainan Island: a retrospective study from 2017–2023", is highly relevant for the epidemiological survey of major helminthiases in a tourist region with a climate favorable to the spread of these neglected parasites.

However, I noticed the absence of information regarding the time of year in which the stool samples were collected—whether during the rainy or predominantly sunny season—and whether participants had recently traveled to another region.

In conclusion, this is an excellent study that underscores the importance of monitoring neglected helminthiases and of implementing effective transmission control measures.

Reviewer #2: Study strengths

• Long 7-year surveillance window on STH dynamics in Hainan Island.

• Large sample (n ≈ 29,669) across multiple counties/cities.

• Species-specific reporting (hookworms, E. vermicularis, A. lumbricoides, T. trichiura).

• Clear temporal trends with year-by-year prevalence and χ² trend tests.

• Spatial analyses identifying clustering and hotspots (Moran’s I, KDE).

• Contextualized with historical data (1986–1991, 2001–2004, 2015).

• Practical relevance for guiding control strategies and policy.

Major concerns and brief recommendations

1) Sampling design and representativeness

- Need explicit site/village selection methods, randomisation/stratification, age distribution (especially 3–10 years), response rates, and any weighting.

2) Diagnostic methodology and case ascertainment

- Specify stool sample count per participant, additional diagnostics, test sensitivity/specificity, infection intensity data, and potential measurement biases.

3) Temporal trends and confounders

- Discuss time-varying confounders (sanitation, deworming programs, economic changes, climate) and, if possible, analyze correlations with these factors.

4) Spatial analysis details

- Provide more methodological specifics: scale/distance thresholds, handling of spatial autocorrelation, validation via simulations, interpretation of LISA clusters, and avoid ecological fallacy.

5) Demography interpretation

- Offer hypotheses for sex/ethnicity/age patterns, consider socioeconomic confounding, and include age-specific or intensity analyses if available.

6 ) Significant changes in prevalence and demographic factors.

Explain the causes of these changes

7) Policy implications

- Add concrete, actionable recommendations (targeted school-based deworming, sanitation improvements, health education) and a monitoring framework with interim targets.

8) Limitations and biases

- Expand to seasonal effects, diagnostic variability, migrant populations, data handling, and propose robust sensitivity analyses.

9) Reproducibility and transparency

- Provide data dictionary, data access plan, and share code or software details and parameters used.

Reviewer #3: 1. The study states that, according to the national surveillance protocol, “two mobile and one fixed sites” were selected to cover 18 counties/cities. However, is the balance between mountainous vs. coastal areas and urban vs. rural settings adequately ensured?

2. The Kato–Katz thick smear method has limited sensitivity for light infections, especially in detecting hookworm and Ascaris lumbricoides, which may lead to underestimation of prevalence. This study did not incorporate molecular methods (e.g., qPCR) or multiple sampling for validation. The reported Kato–Katz detection rate was only 0.38%, whereas the perianal tape method in children yielded 6.47%—a striking discrepancy.

3. Annual infection rates showed a “rebound” in 2020 (from 4.56% in 2019 to 8.72%), yet the authors concluded an overall decline based solely on χ² trend tests. Additional sensitivity analyses or more robust models such as Joinpoint regression or Poisson regression should be applied.

4. Global Moran’s I was significant only for Trichuris trichiura (I = 0.137, P = 0.047), whereas local spatial clustering identified high–high/low–low clusters for multiple species. Could the lack of significance at the global level be due to low overall prevalence and limited statistical power? The study should clarify the choice of spatial weight matrices and apply significance correction (e.g., FDR) to avoid potential false positives.

5. The higher hookworm prevalence was attributed to “barefoot farming,” but the analysis lacks quantitative exploration of other ecological factors such as climate, soil moisture, and behavioral differences among migrant populations.

6. The study reports higher prevalence in Han compared to Li ethnic groups, contrary to historical surveys, yet attributes this difference solely to “improved sanitation in minority regions.” Is there supporting data for this explanation?

7. The authors note the national criterion (<1% prevalence sustained for three consecutive years to indicate transmission control), but prevalence remained >3% during 2021–2023. This suggests Hainan has not yet achieved transmission control. What specific public health strategies (e.g., targeted deworming of key populations, school-based health education, or environmental improvements) should be considered?

8. The study mainly compares its findings with surveillance data from other Chinese provinces, but offers little comparison with international studies (e.g., tropical island countries in Southeast Asia). Such comparisons would strengthen global relevance.

9. The KDE and LISA spatial clustering maps are informative, but the links between spatial patterns and local ecological/economic conditions are insufficiently discussed. For example, why did Wuzhishan emerge as a core high-density zone? Could this be associated with mountainous rainfall, soil pH, or farming practices?

PLOS authors have the option to publish the peer review history of their article (what does this mean? ). If published, this will include your full peer review and any attached files.

**Do you want your identity to be public for this peer review?** For information about this choice, including consent withdrawal, please see our Privacy Policy .

Reviewer #1: **Yes:** Dayane Alvarinho de Oliveira

Reviewer #2: **Yes:** Prof. Ahmed Hassan Fahal

Reviewer #3: No

**Figure resubmission:**
---

## [Decision Letter · Decision Letter 1]

16 Dec 2025

Dear Li,

We are pleased to inform you that your manuscript 'Temporal and spatial distributions and clustering features of soil-transmitted helminthiases on Hainan Island: a retrospective study from 2017–2023' has been provisionally accepted for publication in PLOS Neglected Tropical Diseases.

Best regards,

Krystyna Cwiklinski, PhD

Section Editor

Krystyna Cwiklinski

Section Editor

Shaden Kamhawi

co-Editor-in-Chief

Paul Brindley

co-Editor-in-Chief

The authors have addressed the comments raised by the reviewers. The manuscript is now suitable for publication in PLoS NTD.

Reviewer's Responses to Questions

**Key Review Criteria Required for Acceptance?**

**Methods**

-Are the objectives of the study clearly articulated with a clear testable hypothesis stated?

-Is the study design appropriate to address the stated objectives?

-Is the population clearly described and appropriate for the hypothesis being tested?

-Is the sample size sufficient to ensure adequate power to address the hypothesis being tested?

-Were correct statistical analysis used to support conclusions?

-Are there concerns about ethical or regulatory requirements being met?

Reviewer #3: (No Response)

**Results**

-Does the analysis presented match the analysis plan?

-Are the results clearly and completely presented?

-Are the figures (Tables, Images) of sufficient quality for clarity?

Reviewer #3: (No Response)

**Conclusions**

-Are the conclusions supported by the data presented?

-Are the limitations of analysis clearly described?

-Do the authors discuss how these data can be helpful to advance our understanding of the topic under study?

-Is public health relevance addressed?

Reviewer #3: (No Response)

**Editorial and Data Presentation Modifications?**

Reviewer #3: (No Response)

**Summary and General Comments**

Reviewer #3: (No Response)

PLOS authors have the option to publish the peer review history of their article (what does this mean? ). If published, this will include your full peer review and any attached files.

**Do you want your identity to be public for this peer review?** For information about this choice, including consent withdrawal, please see our Privacy Policy .

Reviewer #3: No

---

## [Editor Report · Acceptance letter]

Dear Li,

We are delighted to inform you that your manuscript, "

Temporal and spatial distributions and clustering features of soil-transmitted helminthiases on Hainan Island: a retrospective study from 2017–2023," has been formally accepted for publication in PLOS Neglected Tropical Diseases.

Best regards,

Shaden Kamhawi

co-Editor-in-Chief

Paul Brindley

co-Editor-in-Chief
